# Improving the Barrier Properties of Paper to Moisture, Air, and Grease with Nanocellulose-Based Coating Suspensions

**DOI:** 10.3390/nano12203675

**Published:** 2022-10-19

**Authors:** André Mazega, Quim Tarrés, Roberto Aguado, Maria Àngels Pèlach, Pere Mutjé, Paulo J. T. Ferreira, Marc Delgado-Aguilar

**Affiliations:** 1LEPAMAP-PRODIS Research Group, University of Girona, C. Maria Aurèlia Capmany, n°61, 17003 Girona, Spain; 2CIEPQPF, Department of Chemical Engineering, University of Coimbra, Pólo II, 3030-790 Coimbra, Portugal

**Keywords:** air resistance, alginate, barrier properties, Kit rating, minerals, nanocellulose, packaging paper, poly(vinyl alcohol), pullulan, water vapor transmission rate

## Abstract

Food packaging manufacturers often resort to lamination, typically with materials which are neither non-biodegradable nor biobased polymers, to confer barrier properties to paper and cardboard. The present work considers a greener solution: enhancing paper’s resistance to moisture, grease, and air by aqueous coating suspensions. For hydrophobization, a combined approach between nanocellulose and common esterifying agents was considered, but the water vapor transmission rate (WVTR) remained excessively high for the goal of wrapping moisture-sensitive products (>600 g m^−2^ d^−1^). Nonetheless, oil-repellant surfaces were effectively obtained with nanocellulose, illite, sodium alginate, and/or poly(vinyl alcohol) (PVA), reaching Kit ratings up to 11. Regarding air resistance, mineral-rich coatings attained values above 1000 Gurley s. In light of these results, nanocellulose, minerals, PVA, pullulan, alginate, and a non-ionic surfactant were combined for multi-purpose coating formulations. It is hypothesized that these materials decrease porosity while complementing each other’s flaws, e.g., PVA succeeds at decreasing porosity but has low dimensional stability. As an example, a suspension mostly constituted by nanocellulose, sizing agents, minerals and PVA yielded a WVTR of roughly 100 g m^−2^ d^−1^, a Kit rating of 12, and an air resistance above 300 s/100 mL. This indicates that multi-purpose coatings can be satisfactorily incorporated into paper structures for food packaging applications, although not as the food contact layer.

## 1. Introduction

Amidst the pressure on packaging manufacturers to replace non-biodegradable plastics, the search for alternatives has become a global trend at both industrial and academic levels [1,2]. This affects, among other materials, paper–polyethylene laminates, which usually contain a barrier layer consisting of ethylene vinyl alcohol (EVOH). In this case, the difficulty of delamination hinders the recyclability and compostability of paper [3], although some peelable laminates have appeared in recent years [4]. Despite the latest efforts of papermakers and researchers, the challenge of attaining paper-based materials with barrier properties that are comparable to those obtained by plastic lamination remains ongoing [5].

This transition involves both materials and production processes. Regarding the former, polysaccharides such as chitosan, alginic acid, and pullulan are prime examples of biobased and biodegradable macromolecules [6,7]. They are highly compatible with paper and effective at repelling grease, since they decrease porosity and are insoluble in non-polar solvents [8]. Nonetheless, their overall barrier properties are allegedly worse than those of poly(vinyl alcohol) (PVA), a synthetic but water-soluble and biodegradable polymer [9]. Desirable materials also include minerals that are found naturally and abundantly in soils and waters. Such is the case of CaCO_3_ and certain clays, which are commonly used in papermaking [10,11]. Another mineral that is involved in this study, sodium tetraborate decahydrate (borax), is naturally occurring and widely used in pesticides. However, as can be inferred from this application, it is not inert [12], and thus only low concentrations are considered. Its capabilities to promote crosslinking between polysaccharide chains may help attain an airtight layer [13].

Regarding processes, there are convincing reasons to prefer conventional paper coating over extrusion coating or lamination, even if the latter is performed with bioplastics [14]. First, the method could be adapted to many paper machines without the need of a lamination line. Second, the need for high temperature (180 °C or more) is avoided. Moreover, papermakers generally exclude organic solvents, and so does the present work. On the one hand, the use of organic solvents to dissolve hydrophobic compounds, otherwise hard to disperse, may achieve excellent protection from moisture. For example, coating paperboard with shellac/ethanol attained a water vapor transmission rate (WVTR), at 25 °C and a relative humidity (RH) of 50%, below 10 g m^−2^ d^−1^ [15]. On the other hand, the use of non-aqueous systems limits applicability on a large scale.

Aqueous suspensions encompassing water-insoluble components can benefit from the presence of cellulose nanofibers (CNFs), both as stabilizer and as rheology modifier [16,17]. Furthermore, they have been extensively used as coating components and are well-known to increase, by themselves, the air resistance of paper by more than one order of magnitude [2]. They can also be partially hydrophobized by esterification with sizing agents, such as alkyl ketene dimer (AKD) and alkenyl succinic anhydride (ASA), commonly used in papermaking [18].

The hypotheses considered in this study can be formulated as follows. First, the addition of AKD and ASA to nanocellulose-based coatings provides protection from water. This hypothesis is not new but quantifying to what extent they contribute to barrier properties is important to define the best plausible approach. Second, the combination of nanocellulose with sodium alginate, PVA, and/or illite, which is a clay mineral unusual in papermaking, improves grease resistance. Third, pullulan, illite, borax, and CaCO_3_ enhance the already-known air barrier properties of nanocellulose coatings. Finally, all these components can be combined in such ways that the resistances to moisture, grease and air are simultaneously improved, and in which resorting to multi-layer coating strategies is not required.

## 2. Materials and Methods

### 2.1. Materials

Bleached eucalyptus kraft pulp (BEKP) was provided by Ence Celulosa y Energía, S.A. (Navia, Spain). 2,2,6,6-tetramethylpiperidine-1-oxyl radical (TEMPO), NaClO (10%, w/v), NaBr, a non-ionic surfactant (Pluronic^®®^ F-127), PVA (Mw 30–70 kDa, 87–90% hydrolyzed), alginic acid sodium salt with viscosity 15–25 cP (1 wt.% in H_2_O), toluene, heptane, castor oil, and NaOH were purchased from Sigma-Aldrich (Schnelldorf, Germany). Calendered uncoated paper of industrial origin with an approximate grammage of 78 g/m^2^, produced from bleached wood kraft pulp, was used in all coating experiments. Powdered illite, paper-grade calcium carbonate powder, AKD, and ASA were also of industrial origin. Pullulan was received from Chem-Lab Analytical (Zedelgem, Belgium).

### 2.2. Production of Nanocellulose

A 30 g amount of BEKP, on a dry weight basis was suspended in water and mixed with NaBr (3.00 g) and TEMPO (0.48 g), after which 335 mL of the aqueous NaClO solution (15 mmol of NaClO per gram of pulp, over dry weight) were then added. The selective oxidation of primary hydroxyl groups took place at 1 wt.% consistency, under gentle stirring (3-blade mechanical stirrer, ~400 rpm) and at 25 °C. NaOH (0.5 M) was added dropwise, all throughout the reaction, to keep the pH at 10. Once the pH was stabilized, the oxidation was considered finished. The resulting oxycellulose pulp was then thoroughly washed with distilled water, vacuum-filtered, and diluted to 3 wt.% consistency. Oxidized fibers were dispersed by means of an Ultra-Turrax device (IKA, model T25) at 16,000 rpm, for 3 min, and ultrasonicated at 35 kHz.

Fibrillation was carried out in a high-pressure homogenizer (HPH) from GEA Niro Soavi (Parma, Italy), model NS1001L2K. A suspension of oxidized fibers was passed 3 times at 300 bar, 3 times at 600 bar, and 3 times at 900 bar. The resulting CNFs had an average degree of polymerization of 197, a carboxylate content of 1526–1640 COO^−^/g, and a transmittance at 600 nm of 85%. Characterization methods are described elsewhere [19,20]. The CNF suspension was stored in plastic bottles and kept at 4 °C to prevent degradation.

### 2.3. Coating Suspensions

Table 1 presents the composition of every coating suspension studied. Formulations were coded with a Greek root for the property that is intended in each case, i.e., resisting moisture (*Hydro*), oil (*Lipo*), or air (*Aero*).

100 mL of the CNF gel were placed into a previously weighted 250-mL borosilicate glass beaker and, if necessary, adjusted to the desired consistency by means of adding water. The suspension was heated to 50 °C under vigorous agitation, causing a gel–sol transition. The other components were added, while distilled water was used to compensate the evaporation losses.

Due to the variety of interactions between the very different and many components of multi-purpose formulations (*Multi*), a small amount of the non-ionic surfactant F-127 was added in some cases to ease interfacial compatibility [21,22]. Such amounts accounted for 0.5% of the total solid content for *Multi 1* and *2*, 0.0% for *Multi 3*, and 2.0% for *Multi 4*.

### 2.4. Surface Treatment

Sheets were placed onto a Mathis laboratory coater, with a pre-drying infrared system coupled to an applicator bar (SVA-IR-B). The effect of two different bars was evaluated, from now on referred to as “smooth roll” and “engraved roll”. In any case, the linear velocity was set as 6 m min^−1^. With the coating suspension still hot, sheets were coated only on one side. Afterwards, they were air-dried at room temperature. To prevent shrinking, paper was physically restrained to a steel plate.

### 2.5. Evaluation of Barrier Properties

The dynamic water contact angle of uncoated and coated sheets was determined along 60 s, using an OCA 15 goniometer (Dataphysics, Germany). WVTR was quantified by the dry cup method [23]. For that, paper samples were placed in impermeable cups containing silica gel. The cups were sealed with an O-ring and kept in a conditioned room at 23 °C and 50% of relative humidity (RH). They were weighed in the same room along 24 h. WVTR can then be calculated from:(1)$$WVTR=\Delta w/\left(A\times t\right)$$
where Δ*w* is the increment in weight until time *t*, and *A* is the transmission area. The weight increase was linear with time for approximately the first 4–6 h (R^2^ > 0.99), and then it tended to level off. Hence, the result of Eq. 1 for the linear section corresponded to the maximum WVTR, while weight increase along 24 h resulted in the 1-day average WVTR.

Kit tests were carried out to evaluate the grease resistance of uncoated and coated sheets, following the TAPPI procedure T559 [24]. The number corresponding to the most aggressive mixture of castor oil, toluene, and heptane that was resisted by paper is reported as the “Kit rating”.

Air resistance was estimated in accordance with the Gurley method, following the ISO standard 5636/5 [25]. Briefly, we measured the time spent for 100 mL of air to pass through a 6.45 cm^2^ cross-sectional area, driven by a pressure gradient of 1.22 kPa.

### 2.6. Other Characterization Techniques

All sheets were weighted and had their thickness measured by means of a digital micrometer. Their bulk density was then calculated as the quotient between their grammage (g/m^2^) and their thickness.

Scanning electron microscopy (SEM) was performed by means of a ZEISS DSM 960A device (ZEISS Iberia, Madrid, Spain) using carbon coating, a secondary electron detector, and a voltage of 5 kV. Cross-sectional views were collected after cryogenic fracture.

An L&W (Munich, Germany) Elrepho spectrophotometer, conforming to ISO 2471 [25], was used to quantify opacity using a C/2° light source. The tensile properties (ISO 1924-2) and the tear index (ISO 1974) were computed by a universal testing machine (Instron, Barcelona, Spain) and by an Elmendorf tester (IDM), respectively [25]. Moreover, internal bond strength was measured with a Scott bond-testing machine from IDM, model IBT 10A, following TAPPI T569 [24].

## 3. Results and Discussion

### 3.1. The Role of Nanocellulose

CNFs played at least four roles in coating processes for barrier properties: thickener, dispersion stabilizer, pore filler, and binder. The first two features made the bar coating process possible, attaining macroscopic homogeneity and sufficient weight gain. The effects of nanocellulose on the initially porous network and its attachment to cellulosic fibers can be appreciated in the SEM images of Figure 1. The entanglement of fibers and fillers in Figure 1a leaves pores that can be more easily appreciated in the inset figure. Nonetheless, placing CNFs onto this surface apparently either sealed or decreased the size of most of the pores (Figure 1b). The cross-sectional view reveals a new thin layer over the sheet, but nanofibers that penetrated paper were necessary to bind it to the inner fibers (inset figure). Regarding Figure 1c, representing a sample with ASA and AKD (*Hydro 3*), there are effects on the fiber surface that are not found in conventionally ASA-sized or AKD-sized papers [26]. Likely, the hydrophobization of CNFs with these sizing agents promoted their aggregation in aqueous media, forming flocs over the fibers. In another context, when nanocellulose was combined with water-soluble polymers, as in Figure 1d (*Multi 2*), the decrease in porosity (surface view) and the formation of a distinguishable layer (cross-sectional view) were clearer than those of *CNF2%* (Figure 1b).

Nonetheless, the thickness of the thin layer that can be observed in Figure 1b is much lower than the total increase in thickness, as can be seen from Table 2. Indeed, not all CNFs remained on the surface. Many of them penetrated the sheet transversally, established hydrogen bonds with inner fibers, and increased the amount of bound water. As a result, after drying, the paper became considerably thicker. Another proof lies in the internal bonding strength, as coating with nanocellulose suspensions (2 wt.%) increased it from 0.5 to 0.7 kJ m^−2^. Since CNFs partially penetrated the sheet, they offered more surfaces for intermolecular interactions. This enabled paper to absorb more energy before rupturing. However, its performance in tensile tests was not enhanced, as effects on the breaking length were either non-significant or detrimental (Table 2). Detrimental effects can be attributed to the aforementioned process of wetting and re-drying [27].

It should be noted that, before regarding CNFs as a food contact material, the uncertainty surrounding the potential effects of nanofibrillated cellulose in the small intestine has yet to be addressed [28]. Considering the current state of the art and current regulations, these nanocellulose-based coatings should not constitute the inner layer in direct contact with liquids or with wet foodstuff [29]. Instead, they are suggested as outer or intermediate barrier layers within multilayer systems that are free of non-biodegradable polymers.

### 3.2. Enhancement of Water Barrier Properties

Owing to the hydrophobic alkyl chains, the surface of paper sheets with AKD and ASA was consistently more hydrophobic than that of uncoated paper, as inferred from Figure 2. Differences between the two applicators were non-significant, and thus only those of the engraved roll are shown. As described in other works [30,31], CNFs reacted with the sizing agents. This is often assumed to generate ester bonds by alcoholysis in the case of ASA, and β-keto ester bonds in the case of AKD [31]. Alternatively, the carbonyl oxygen of AKD can act as hydrogen bond acceptor towards cellulose. In any case, equatorial planes (including –OH and –COO^−^ groups) became less available for hydrogen bonding with water molecules, due to part of them being attached to alkyl chains. Hence, water–water interactions were entropically favored over water–surface interactions. In addition, nanocellulose also stabilized the aqueous dispersion of the hydrophobic by-products that resulted from the hydrolysis of ASA and AKD.

The greatest initial water contact angle was obtained when using the highest proportion of ASA, 0.75%, and the wetting speed was found to be slow (*Hydro 3*). The inset picture in Figure 2 highlights the hydrophobic behavior of this formulation. Nonetheless, the composition *Hydro 2*, where the CNF content accounts for 1.25%, did not maintain the high contact angle of the first two seconds. In any case, all these combinations of AKD and ASA yielded contact angle values in the same order of magnitude, as if using them separately [32].

When using only CNFs for comparison purposes (*CNF2*%), targeting the same solid content, the dynamic water contact angle was much lower than that of the reference paper sheets. The decrease in the initial contact angle (from 95° to 79°) and the increase in the wetting speed (from 0.02 to 0.4 °/s) have two plausible reasons. First, the disposition of CNFs physically changed the topography of the paper surface, and thus the liquid–solid and air–solid surface tension. Second, the hydrophilicity of CNFs, especially considering their carboxylate groups, resulted in higher liquid–solid adhesive forces.

Even though the water contact angle was in the typical range of hydrophobic materials (>90°) [33], packaging paper did not succeed at blocking water vapor diffusion. The values in Figure 3 for the maximum WVTR (first 4 h) and the 1-day average WVTR (engraved roll) are in the same order of magnitude as those of uncoated paper. On the one hand, comparatively speaking, the maximum rate of vapor flow decreased by up to 35% from the control experiment, even when using low amounts of AKD and ASA. On the other hand, these values are not low enough to protect moisture-sensitive products, including most foodstuffs. Other kinds of paper coating, e.g., those based on resins or waxes, have attained WVTR values below 300 g m^−2^ d^−2^ with a similar coat weight [34,35].

Figure 3 also shows little difference between the engraved roll and the smooth one. In fact, the coat weight (~2 g m^−2^) was similar in both cases, as can be deduced from the grammage values in Table 3. One-way ANOVA tests revealed that the kind of roll exerted no significant influence on the coat weight applied (*p* = 0.63), on its thickness (*p* = 0.55), on the bulk density (*p* = 0.77), or on breaking length (*p* = 0.89).

### 3.3. Improvement in Grease Resistance

Uncoated paper could not be considered grease-proof since its Kit rating was 0, as shown in Figure 4. In other words, it wicked pure castor oil. When the surface was treated with oxidized CNFs (3 wt.%), it only failed to resist oil flow insofar as enough toluene and heptane (80–90 wt.%) were incorporated. Detrimental synergetic effects between the different components caused certain combinations (*Lipo*
*1*, *6*, *7*) to perform worse than nanocellulose alone. In contrast, a Kit rating of 11 was attained by compositions *Lipo 4* and *Lipo 5*, characterized by its high content of illite and sodium alginate, respectively. This indicates that the sheet resisted a 1:1 (*v*/*v*) mixture of these organic solvents. Moreover, the engraved roll outperformed the smooth one in these cases.

However, it could not be said that one component performed better than any other. The *Lipo 8* formulation, which obtained a Kit rating of 10, comprised neither illite nor Alg-Na. The *Lipo 2* formulation, which also passed the ninth or the tenth Kit ratings, encompassed all components (nanocellulose, Alg-Na, illite, PVA). If anything, it is worth noting that the proportion of CNFs in *Lipo 2* is the lowest (62.5% with respect to total solids). Therefore, the only feature that the best formulations for grease resistance had in common is an extreme value for at least one of the components. In any case, it can be stated that every one of them, including nanocellulose, exerted a positive influence on the ability of paper to repel oils. In other words, all of them contributed to sealing the pores of paper [8]. No less importantly, they did not become solvated by oil or non-polar solvents, and their intermolecular interactions with oils (e.g., dispersive forces) did not replace their intense hydrogen bonding.

Similarly, the texture of the applicator roll exerted no significant influence on the coat weight (*p* = 0.93), on its thickness (*p* = 0.83), or on the breaking length (*p* = 0.28), as inferred from ANOVA tests performed on the data of Table 4. Under a flexible criterion (*α* = 0.1), the difference in density is significant (*p* = 0.07) and it can be stated that the smooth roll attained bulkier sheets.

### 3.4. Gas Barrier Properties

CNFs, both alone and combined with inorganic minerals and/or pullulan, increased the air resistance of paper sheets (9.4 s/100 mL for uncoated paper) by roughly two orders of magnitude, as displayed in Figure 5. Nonetheless, at least when using the smooth applicator, all-polymeric coatings, with nanocellulose and pullulan (*Aero 2*), performed significantly worse than formulations comprising minerals. The most successful formulations were those with calcium carbonate (*Aero 4* and *5*). Beneficial flocculation effects by using CNFs with CaCO_3_ have been found in previous works [36], albeit in the context of the wet end section of the paper machine, instead of the coating section.

Overall, the mechanism of air blocking can be described in two ways. First, the crystalline structure of the minerals and the highly hydrated nanocellulose, generally with interatomic distances below 300 pm [37,38], blocked the intraparticle diffusion of N_2_ and O_2_. Second, the small size of crystallites, held together by CNFs as a stabilizer, also hindered interparticle diffusion by attaining high tortuosity.

Unlike in the previous two subsections, intermolecular interactions can be safely neglected in the case of air with the components of *Aero 1–5*. Hence, diffusion through the sheet is likely governed by a Knudsen mechanism, rather than by molecular diffusion, considering the slenderness of interparticle pores. In Knudsen diffusion, the transport coefficient is inversely proportional to tortuosity [39]. The engraved roll, although attaining higher coat weights in most cases (Table 5), exerted either a non-significant influence or a detrimental effect on air resistance, given that it might have promoted preferential pathways.

As found for hydrophobic and lipophilic formulations, the type of roll did not attain significant differences upon weight, thickness, or breaking length (*p* > 0.05). In another context, application with the smooth roll imparted a significantly higher increase in the average values of the internal cohesion (*p* = 0.03), but the magnitude of this increase was lower than the mean interval length. In any case, all nanocellulose-based suspensions enhanced this absorption of energy before rupture by 26–44%. Since the presence of other components exerted no significant effects, it can be stated that the incorporation of CNFs onto and into the sheet, promoting hydrogen bonding across the Z direction, was the main cause of this improvement.

### 3.5. Multi-Purpose Barrier Properties

With few exceptions, the formulations *Multi 1–4* had positive effects on the properties displayed in Figure 6. Only one of these properties, the dynamic contact angle (Figure 6a), neither matched nor surpassed the performance of single-objective coatings. The carbohydrate polymers added succeeded at hindering diffusion through the sheet but provided water–polymer hydrogen bonding capability [40]. Consequently, they compensated the hydrophobic effect of AKD and ASA. The lowest surface wettability was attained when reducing the percentage of CNFs to 1%, as in *Hydro 3*.

Wettability and WVTR (Figure 6b) were not correlated. The best result was attained by *Multi 2*, where the transmission rate was roughly 15 times lower than that of uncoated paper. In this case, nearly half of the total content of solids amounted to PVA. Even though it is a polyol, its crystallinity ensures that its intra- and intermolecular hydrogen bonds are not replaced with water [41]. Furthermore, its homogeneous distribution avoids preferential pathways. Interestingly, the WVTR of PVA films at 23 °C and 50% RH may be negligible [42].

In a development pertaining to oil barrier properties, both *Multi 1* (PVA-free) and *Multi 2* passed the highest Kit rating test (Figure 6c). As described above, nanocellulose alone had attained high grease resistance (8–9). This was further enhanced (to 10–11) by pullulan and minerals. Then, the resistance of these systems to oil was enhanced by adding water-soluble components such as sodium alginate. A homogeneous application was key to avoid non-blocked paths, given that in the case of *Multi 3*, with no surfactant to grant good dispersion, samples failed the fifth Kit rating. Finally, the low CNF content (1%) of *Multi 4*, along with the low coat weight (Table 6), explains why this composition did not reach the greaseproof performance of *Multi 1–2*.

All multi-purpose coating formulations increased air resistance by at least two orders of magnitude (Figure 6d). Nonetheless, *Multi 1* and *Multi 4* performed better than the other two formulations. *Multi 3*′s failure to reach a resistance of 1000 Gurley s could be explained again by a heterogeneous distribution, leaving preferential channels for interparticle diffusion. *Multi 2* did not reach that value either, but the air resistance attained lies in the same order of magnitude as that of PVA-sized paper [43].

Since only the engraved roll was used for multi-purpose suspensions, Table 6 does not compare different applicators. Instead, other relevant properties of packaging paper are shown. This is to highlight, for instance, that this coating method mostly preserved the original mechanical properties of paper. The breaking length and the tear index even increased in the case of the surfactant-free composition. In other cases, the presence of the surfactant was non-significant or detrimental to tensile properties, as it may weaken inter-fiber bonds [44,45].

Similar to airtight formulations, the internal bond strength was significantly enhanced from uncoated sheets to papers coated with multi-objective suspensions (Table 6). For *Multi 1* and *Multi 3*, where the percentage of CNFs was 2.5 wt.%, there was no delamination at all. *Multi 2* had the same concentration of CNFs, but the lower compatibility of PVA might have eased rupture. As a final remark, the opacity, which can be considered a barrier property as well (i.e., to light), hardly underwent changes. Despite the presence of white, opaque minerals, the main components were highly transparent CNFs and PVA.

## 4. Conclusions

Coating paper sheets with CNFs effectively decreased paper’s oil-wicking ability, increasing the Kit rating from 0 to 8–9. It also improved air resistance by two orders of magnitude. As a drawback, the surface became more hydrophilic than that of the uncoated sheets. Incorporating two common paper-grade sizing agents, AKD (0.25 wt.%) and ASA (0.75 wt.%), resulted in a dynamic water contact angle of 112–121° during the first 60 s of wetting. However, the maximum WVTR was only reduced by 35%. Interestingly, the formulations intentionally designed for multi-objective barrier properties, containing pullulan, sodium alginate, PVA, and minerals, lowered WVTR by one order of magnitude. When the PVA concentration was 4%, this reduction was from 1547 to 104 g m^−2^ d^−1^. Similarly, some multi-objective formulations (*Multi 1* and *Multi 2*) passed even the most aggressive tests for grease resistance. Nevertheless, they only increased slightly, or even decreased, the water contact angle, and did not surpass nanocellulose/CaCO_3_ coatings for airproof purposes.

All considered, none of the combinations exceled at every aspect. *Multi 2*-coated papers were resistant to water vapor transmission and oil wicking, but suboptimal in terms of air resistance (<400 Gurley s) and water contact angle (<100°). *Multi 4*, in contrast, attained air resistance values over 2000 s/100 cm^3^, but failed the ninth Kit test. Therefore, the recommendation of a certain coating composition depends on the application and the specific standards to meet.

## Figures and Tables

**Figure 1 nanomaterials-12-03675-f001:**
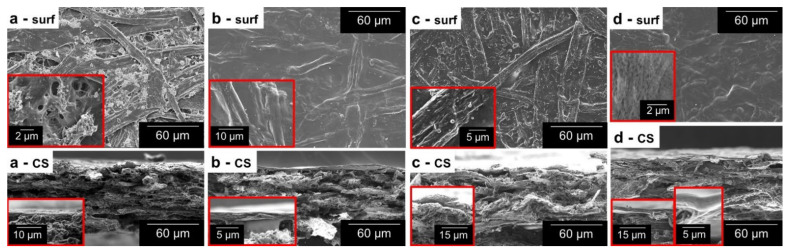
Micrographs of the surface (*surf*) and the cross-section (*CS*) of uncoated paper (**a**), *CNF2%* (**b**), *Hydro 3* (**c**), *and Multi 2* (**d**). Inset images display magnified sections of the same sample.

**Figure 2 nanomaterials-12-03675-f002:**
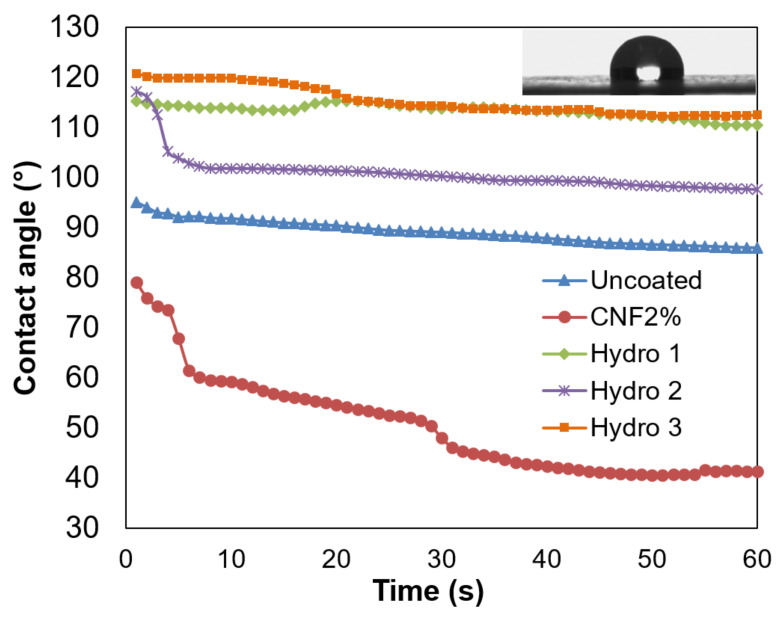
Dynamic contact angle of uncoated paper and sheets that were coated with formulations *Hydro 1*–*3*, using the engraved applicator roll. The inset image corresponds to a water drop over a paper sheet coated with *Hydro 3*.

**Figure 3 nanomaterials-12-03675-f003:**
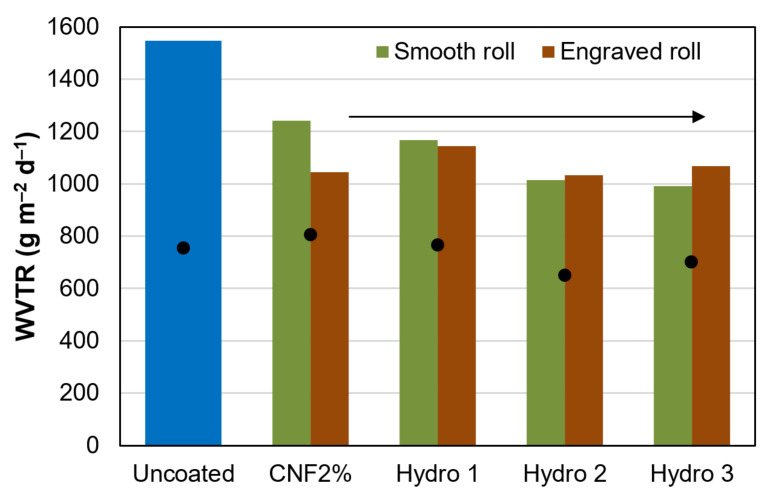
Maximum (columns) and 24-h average (dots) water vapor transmission rate through uncoated paper and through sheets that were coated with formulations *Hydro 1–3*.

**Figure 4 nanomaterials-12-03675-f004:**
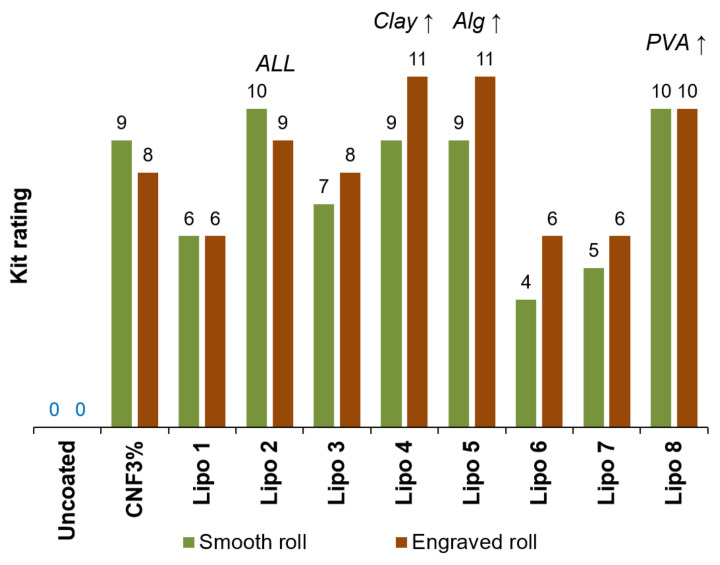
Results of the Kit test (anti-wicking of oil) for uncoated paper and paper that was coated with formulations *Lipo 1–8*.

**Figure 5 nanomaterials-12-03675-f005:**
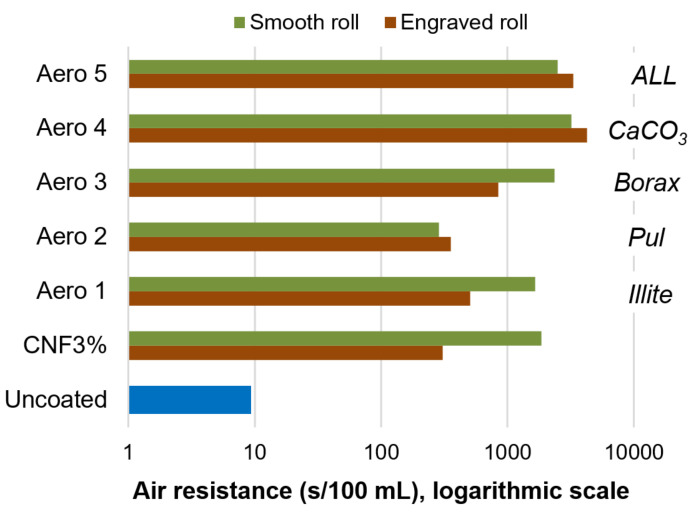
Air resistance (Gurley method) for uncoated paper, nanocellulose-coated paper, and sheets that were coated with formulations *Aero 1–5*.

**Figure 6 nanomaterials-12-03675-f006:**
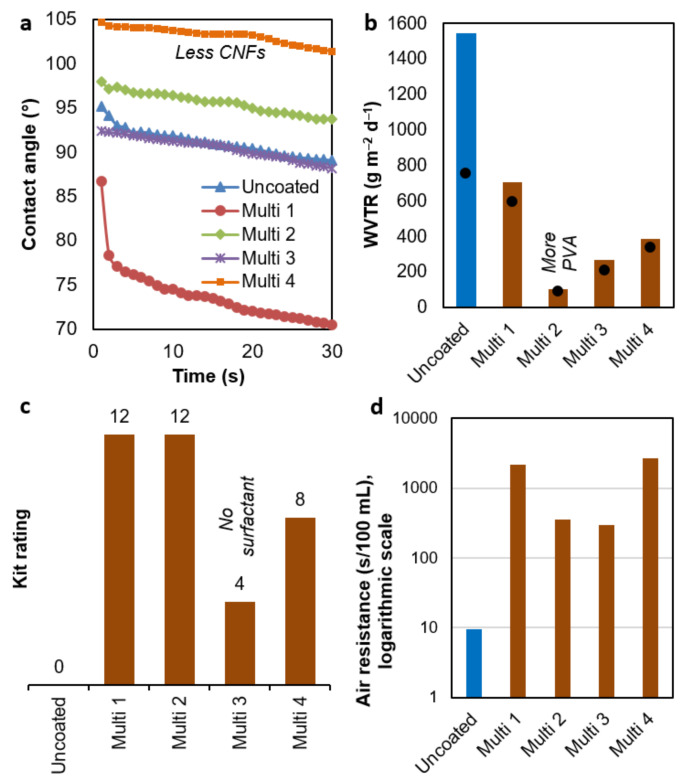
Key barrier properties of sheets coated with multi-purpose suspensions: (**a**) dynamic contact angle; (**b**) maximum WVTR (columns), 24 h average WVTR (dots); (**c**) results of greaseproof tests, and (**d**) air resistance (Gurley method). The engraved roll was used in all cases.

**Table 1 nanomaterials-12-03675-t001:** Coding and composition of each of the aqueous coating formulations, classified according to their purpose.

Coating	CNFs (wt.%)	AKD (wt.%)	ASA (wt.%)	Alg-Na (wt.%)	Illite (wt.%)	PVA (wt.%)	Pul (wt.%)	Borax (wt.%)	CaCO_3_ (wt.%)
Uncoated	--	--	--	--	--	--	--	--	--
Nanocellulose-only formulations:									
CNF2%	2.00	--	--	--	--	--	--	--	--
CNF3%	3.00	--	--	--	--	--	--	--	--
For water barrier properties:									
Hydro 1	1.50	0.25	0.25	--	--	--	--	--	--
Hydro 2	1.25	0.25	0.50	--	--	--	--	--	--
Hydro 3	1.00	0.25	0.75	--	--	--	--	--	--
For grease resistance:									
Lipo 1	2.18	--	--	0.23	0.60	--	--	--	--
Lipo 2	1.88	--	--	0.23	0.60	0.30	--	--	--
Lipo 3	2.44	--	--	0.11	0.30	0.15	--	--	--
Lipo 4	2.40	--	--	--	0.60	--	--	--	--
Lipo 5	2.78	--	--	0.23	--	--	--	--	--
Lipo 6	2.10	--	--	--	0.60	0.30	--	--	--
Lipo 7	2.48	--	--	0.23	--	0.30	--	--	--
Lipo 8	2.70	--	--	--	--	0.30	--	--	--
For gas-blocking behavior:									
Aero 1	2.40	--	--	--	0.60	--	--	--	--
Aero 2	2.94	--	--	--	--	--	0.06	--	--
Aero 3	2.98	--	--	--	--	--	--	0.02	--
Aero 4	2.98	--	--	--	--	--	--	--	0.02
Aero 5	2.30	--	--	--	0.60	--	0.06	0.02	0.02
For multi-purpose barrier properties:									
Multi 1	2.50	0.25	0.75	0.25	0.60	--	0.08	0.03	0.03
Multi 2	2.50	0.25	0.75	0.25	0.60	4.00	0.08	0.03	0.03
Multi 3	2.50	0.25	0.75	0.25	0.60	3.00	0.08	0.03	0.03
Multi 4	1.00	0.25	0.75	0.25	0.60	3.00	0.08	0.03	0.03

Alg-Na: sodium alginate. Pul: pullulan. All percentages are expressed on the basis of the wet weight of the coating suspension.

**Table 2 nanomaterials-12-03675-t002:** Weight, dimensional, and mechanical properties of the sheets coated only with aqueous suspensions of cellulose nanofibers. The amplitude of the intervals equals twice the standard deviation.

Applicator Roll	Coating	Grammage (g m^−2^)	Thickness (μm)	Density (g cm^−3^)	Breaking Length (km)	Internal Bond Strength (kJ m^−2^)
None	Uncoated	78.5 ± 0.4	90 ± 3	0.87 ± 0.02	4.75 ± 0.05	0.50 ± 0.01
Engraved	CNF2%	80.8 ± 0.2	102 ± 1	0.79	4.6 ± 0.1	0.71
	CNF3%	83.5 ± 0.3	104 ± 1	0.80 ± 0.01	4.41 ± 0.04	0.70 ± 0.07
Smooth	CNF2%	80.0	103 ± 1	0.77 ± 0.01	4.8 ± 0.3	0.69 ± 0.02
	CNF3%	83.7 ± 0.6	104 ± 1	0.81	4.7 ± 0.2	0.67 ± 0.02

**Table 3 nanomaterials-12-03675-t003:** Key properties of the sheets coated with hydrophobic formulations. The amplitude of the intervals equals twice the standard deviation.

Applicator Roll	Coating Formulation	Grammage (g m^−2^)	Thickness (μm)	Density (g cm^−3^)	Breaking Length (km)
None	Uncoated	78.5 ± 0.4	90 ± 3	0.87 ± 0.02	4.75 ± 0.05
Engraved	Hydro 1	79.9 ± 0.3	108 ± 4	0.74 ± 0.03	4.6 ± 0.1
	Hydro 2	80.7 ± 0.3	102 ± 1	0.79 ± 0.01	4.6 ± 0.4
	Hydro 3	79.8 ± 0.1	105 ± 3	0.76 ± 0.02	4.9 ± 0.2
Smooth	Hydro 1	79.9 ± 0.5	102 ± 2	0.79 ± 0.01	4.66 ± 0.05
	Hydro 2	80.2 ± 0.1	103 ± 3	0.78 ± 0.02	4.6 ± 0.1
	Hydro 3	80.5 ± 0.2	105 ± 1	0.76	4.75 ± 0.04

**Table 4 nanomaterials-12-03675-t004:** Key properties of the sheets coated with lipophobic formulations. The amplitude of the intervals equals twice the standard deviation.

Applicator Roll	Coating Formulation	Grammage (g m^−2^)	Thickness (μm)	Density (g cm^−3^)	Breaking Length (km)
None	Uncoated	78.5 ± 0.4	90 ± 3	0.87 ± 0.02	4.75 ± 0.05
Engraved	Lipo 1	82.4 ± 0.3	105	0.78	4.9 ± 0.2
	Lipo 2	82.9 ± 1.5	109 ± 7	0.76 ± 0.06	4.48 ± 0.02
	Lipo 3	84.8 ± 0.8	112 ± 1	0.75 ± 0.01	4.6 ± 0.5
	Lipo 4	82.2 ± 0.4	104 ± 6	0.79 ± 0.05	4.4 ± 0.1
	Lipo 5	84.5 ± 0.4	108 ± 4	0.79 ± 0.03	4.6 ± 0.2
	Lipo 6	84.3 ± 0.6	108 ± 4	0.79 ± 0.04	4.6 ± 0.1
	Lipo 7	81.6 ± 0.5	106 ± 2	0.77 ± 0.02	4.9 ± 0.3
	Lipo 8	83.1 ± 0.1	106 ± 1	0.78 ± 0.01	4.65 ± 0.02
Smooth	Lipo 1	83.7	102 ± 1	0.82 ± 0.01	4.46 ± 0.02
	Lipo 2	83.0 ± 0.3	102 ± 3	0.81 ± 0.03	4.9 ± 0.2
	Lipo 3	83.2 ± 0.4	100 ± 1	0.83 ± 0.01	4.72 ± 0.03
	Lipo 4	81.9 ± 0.3	108 ± 5	0.76 ± 0.03	4.6 ± 0.3
	Lipo 5	82.4 ± 0.1	111	0.74	4.71 ± 0.06
	Lipo 6	82.2 ± 0.6	104 ± 1	0.79 ± 0.01	4.94 ± 0.08
	Lipo 7	81.7 ± 0.2	103 ± 3	0.79 ± 0.02	5.0 ± 0.1
	Lipo 8	82.1 ± 0.3	105 ± 1	0.78	4.57 ± 0.02

**Table 5 nanomaterials-12-03675-t005:** Key properties of the sheets coated with airtight formulations. The amplitude of the intervals equals twice the standard deviation.

Applicator Roll	Coating Formulation	Grammage (g m^−2^)	Thickness (μm)	Density (g cm^−3^)	Breaking Length (km)	Internal Bond Strength (kJ m^−2^)
None	Uncoated	78.5 ± 0.4	90 ± 3	0.87 ± 0.02	4.75 ± 0.05	0.50 ± 0.01
Engraved	Aero 1	82.0 ± 0.1	105	0.78	4.41 ± 0.04	0.67 ± 0.03
	Aero 2	81.8 ± 0.1	108 ± 1	0.75	4.6 ± 0.2	0.67 ± 0.04
	Aero 3	82.6 ± 0.6	109	0.76 ± 0.01	4.3 ± 0.3	0.68 ± 0.02
	Aero 4	82.3 ± 0.2	107 ± 1	0.77 ± 0.01	4.50 ± 0.09	0.63 ± 0.08
	Aero 5	82.8 ± 0.3	110 ± 1	0.75 ± 0.01	4.7 ± 0.2	0.67 ± 0.02
Smooth	Aero 1	82.5 ± 0.4	109	0.76	4.56 ± 0.08	0.72 ± 0.04
	Aero 2	81.2 ± 0.4	107	0.76	4.5 ± 0.1	0.72 ± 0.07
	Aero 3	82.4 ± 0.3	107 ± 1	0.77 ± 0.01	4.3	0.69 ± 0.05
	Aero 4	82.2 ± 0.3	109 ± 1	0.75 ± 0.01	4.4 ± 0.2	0.68 ± 0.03
	Aero 5	82.2 ± 0.2	108 ± 2	0.76 ± 0.02	4.65 ± 0.01	No rupture

**Table 6 nanomaterials-12-03675-t006:** Weight, dimensional, optical, and mechanical properties of the sheets coated with multi-purpose formulations. The amplitude of the intervals equals twice the standard deviation.

Coating	Grammage (g m^−2^)	Thickness (μm)	Density (g cm^−3^)	Opacity (%)	Breaking Length (km)	Tear Index (N m^2^ kg^−1^)	Internal Bond Strength (kJ m^−2^)
Uncoated	78.5 ± 0.4	90 ± 3	0.87 ± 0.02	82.9	4.75 ± 0.05	7.8 ± 0.1	0.50 ± 0.01
Multi 1	83.5 ± 0.2	110 ± 1	0.76 ± 0.01	83.8	4.2 ± 0.2	8.0 ± 0.4	No rupture
Multi 2	85.5 ± 0.3	110	0.78	83.8	4.63 ± 0.09	8.5 ± 0.1	0.70 ± 0.05
Multi 3	87.8 ± 0.5	111	0.79	82.3	5.06 ± 0.02	7.7 ± 0.1	No rupture
Multi 4	81.8 ± 0.1	107 ± 1	0.76 ± 0.01	84.0	4.70 ± 0.4	8.2 ± 0.2	0.64 ± 0.03

## Data Availability

Available at the digital repository of the University of Girona, http://dugi.udg.edu (accessed on 16 October 2022).

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
