# Peer review of "Improving the Barrier Properties of Paper to Moisture, Air, and Grease with Nanocellulose-Based Coating Suspensions"

_nanomaterials, 2022, doi:10.3390/nano12203675_

Round 1

Reviewer 1 Report

First I have to state that my expertise is rather far removed from such technology. Please take this into account when considering my evaluation.

I read the manuscript with interest. The work described seemed to me systematic, performed with high competence and empiric, as such optimization normally is. The manuscript is well written in good language.

I miss a description of the Ultra-Turrax device (line 95) and of what is the “highest numbered mixture” (line 137).

My field of competence is in food safety. This is not addressed at all. I miss it, as it is not possible to develop technologies with taking risk assessment into account. The closer the contact with food, the higher tends to be the migration, which either calls for thorough risk assessment or restricts the use of compounds and reactions. Nanofibers may not migrate into dry foods (except by abrasion), but with liquid contact this becomes a subject to be well assessed. I’m aware that the assessment of the safety of paper and board is anyway far from what is needed for to comply with legal safety requirements, but further treatment and closer contact aggravates the problem. Most of the additives proposed might be of little concern, but the nanofibers and their treatment clearly are of potential concern. It is a worrying fact that paper and board are much less investigated for safety than the plastics they should replace.

I would like to see some reflections on this, perhaps in a discussion or the conclusion.

Author Response

As the corresponding author, and being well aware that cellulose nanofibers are not regarded as FCMs by the EFSA (EU), at least not yet, I highly appreciate these valuable comments. Significant migration to wet foodstuff or food simulants is 100% sure to happen, and right now, for this packaging to be legally commercialized in the EU (where the authors are), the coatings we present should work as an outer layer, or be covered by yet another layer. Out of caution, at no moment do we presume that this layer is to be in contact with food, but we are explicit with this in the revised version.

Modification in the abstract: "This indicates that multi-purpose coatings can be satisfactorily incorporated into paper structures for food packaging applications, although not as the food contact layer."

In the new section "3.1", introduced to focus particularly on nanocellulose:

"It should be noted that, before regarding CNFs as a food contact material, the uncertainty surrounding the potential effects of nanofibrillated cellulose in the small intestine has yet to be addressed [25]. Considering the current state of the art and current regulations, these nanocellulose-based coatings should not constitute the inner layer in direct contact with liquids or with wet foodstuff [26]. Instead, they are suggested as outer or intermediate barrier layers within multilayer systems that are free of non-biodegradable polymers."

Regarding the punctual queries:

The Ultra-Turrax device is a T25 model from IKA.

"The highest numbered mixture" -> "The number corresponding to the most aggressive mixture". This can be checked in the standard itself (T559), which assigns numbers from 1 to 12 to different compositions.

Reviewer 2 Report

In this manuscript, biodegradable macromolecules include nanocellulose, pullulan, alginate, minerals, poly (vinyl alcohol), and sizing agents were used to improve paper barrier properties by coating. The technologies used in this study are very advanced and the materials used are biodegradable. This research is very meaningful and provides a theoretical basis for the study of multipurpose coatings that can be incorporated into paper structures for food packaging applications. However, the authors should address the following comment.

1.        In lines 15-20, could you explain more clearly the complementary relationship between the selected raw materials?

2.        In line 20, please check whether the logical relationship of “Finally” in the article is reasonable.

3.        In line 35, please check whether the logical relationship of “However” in the article is reasonable.

4.        In line 46, could you explain more clearly the advantages of borax in this experiment?

5.        In line 119, it is suggested that the basis for the addition of each substance be detailed.

6.        In line 171, it is suggested that the mechanism of change in this indicator could be explained and comparisons with other papers added.

7.        In line 206, “0.71 kJ m–2” is not mentioned in figures and tables.

8.        In section 3.2, in the description of the results of “Kit test”, the chart used “Kit number”, while the abstract and body used “Kit rating”, the expression is not uniform.

9.        It is recommended to supplement the mechanism of action leading to the change of the indicator in section 3.2.

10.    It is recommended to unify the positions of “Engraved roll” and “Smooth roll” in Figure 2 and Figure 3.

11.    In line 302 and 317, please check whether the (Fig. 4c) and (Fig. 4d) representations are correct.

12.    In line 344, (0.75 wt%) was not consistent with the previous study.

13.    Table 3 missing header on page 8.

Author Response

We thank the reviewer for the recommendations and acknowledge their attention to detail, observing relevant aspects that we simply skipped (unintentionally) during the preparation of the manuscript. Here follows a point-by-point response.

The attachment contains the same information as this response, but it may be easier to follow as the reviewer's comments have been pasted and different colors are used. 

1. For example, PVA alone would suffice to attain good barrier properties, but it wears off easily if wet, its viscosity is not enough for coating, and it has low dimensional stability. Polysaccharides improve the film-forming properties and the stability, and have been traditionally good for repelling grease. Nanocellulose works as a stabilizer and thickener without defeating the purpose of decreasing porosity. CaCO3 and illite avoid intraparticle diffusion and also help attain proper viscosity ranges. Borax crosslinks polysaccharides. Nonetheless, we have to be concise in the abstract, so this has been summarized as:

"It is hypothesized that these materials decrease porosity while complementing each other’s flaws; e.g., PVA succeeds at decreasing porosity but has low dimensional stability. "

Kindly note that it is logically followed by the following sentence: "As an example, a suspension mostly constituted by nanocellulose, sizing agents, minerals and PVA..."

2. "Finally" -> "In light of these results"

3. "However" -> "Despite the latest efforts of papermakers and researchers". We believed that the revised connecting phrase introduces the main clause in a better way while being logically related to the previous sentence ("recent years"..."latest efforts").

4. Borax has been previously used as a crosslinker for oxycellulose, poly(acrylic acid), and different polysaccharides. For instance, https://www.sciencedirect.com/science/article/pii/S0141813021023722

We now cite this as justification. "Its capabilities to promote crosslinking between polysaccharide chains may help attain an airtight layer [12]."

5. This comment can be addressed either by understanding "basis" as "motivation" or by clarifying the weight basis for the percentages.

New table footnote: "All percentages are expressed on the basis of the wet weight of the coating suspension."

Nonetheless, the addition of surfactant is expressed on the basis of the dry weight or solid content — this was already specified in the original submission. After all, even for Multi 4, where it accounts for 0.12% of the wet weight, it is below the critical micelle concentration (3.3 g/L, 10.3144/expresspolymlett.2016.20). Therefore, its ratio to the components to get associated with is more relevant than its concentration in the aqueous medium.

Regarding motivation, we believe that the place for this is the introduction, which has been enriched.

6. Most likely, the mechanism was described too concisely, but the original submission already highlighted the entropically favored water-water H-bonding over surface-water. Nonetheless, this has been enriched and clarified. For instance, instead of only saying "less equatorial planes were available", we clarify that we refer to hydroxyl and carboxylate groups. Not being a cellulose-specific journal, it is expected that many readers may not have a deep notion of the stereochemistry of cellulose. Plus, we clarify that they became less available "due to being attached to alkyl chains".

Another reference has been added for comparison purposes. It is worth showing that combining AKD and ASA is not necessary:

"In any case, all these combinations of AKD and ASA yielded contact angle values in the same order of magnitude as if using them separately [30]."

7. Please check the new subsection on nanocellulose alone, 3.1. This has been introduced in the revised version to address several comments from all reviewers. Data on CNF2% and CNF3%, using both rolls, have been incorporated into the new Table 2. Consequently, the discussion related to it has also been moved to that part of the manuscript.

8. Uniformized to "Kit rating", which is the term that the standard uses for this result.

9. Briefly, the mechanism consists of sealing the pores of paper with substances that are: a) solid, so molecules do not move one past another; b) insoluble in oil, and c) insoluble in non-polar solvents.

"In other words, all of them contributed to sealing the pores of paper [8]. Not less importantly, they do not become solvated by oil or non-polar solvents, and their intermolecular interactions with oils (e.g., dispersive forces) do not replace their intense hydrogen bonding.

10. Since Figures 3 and 4 were already consistent with each other, we have changed Figure 2 accordingly.

11. The vertical alignment has been corrected and the nomenclature has been changed at the request of another reviewer. Other than that, we see no further issues.

12. The "wt%" mistake, inconsistent with the rest of the manuscript, has been corrected to "wt.%". Besides this, we see no further issues — 0.25% AKD, 0.75% ASA, wet weight basis, is the specification for Hydro 3 (Figure 1, orange squares). 

13. Header issue solved.

Reviewer 3 Report

The work titled "Nanocellulose, pullulan, alginate, minerals, poly(vinyl alcohol) and sizing agents: Improving paper barrier properties by coating" requirres some minor corrections before publication.

What was the type of paper?

line 44 and 80 - subscript

Table 1. I suggest using abbrev. "Multi" instead of "Poly", the same changes in the text.

Point 2.5. Due to lot of wrong calculations of WVTR in many articles I suggest adding equation for WVTR determination.

Figure 1. Reference paper should be also presented here.

Page 6 - Were any differences in contact angle values for engraved and smooth roll? It should be included in the disscusion too.

Page 9 - roughness should be analytically confirmed via e.g. SEM (surface and cross-section)

Line 297 - There is no information about polymer crustallinity, add some reference at least

I suggest using therm "grammage" instead of "basis weight"

Due to the manuscript was submitted to "Nanomaterials" Journal role of nanocellulose should be highlighted in the study.

Author Response

We gratefully take the opportunity to submit an improved version of our manuscript. The reviewer's constructive criticism helped us in this task. Kindly note that this version implies additional experimental work, namely for the new section "3.1", which includes SEM images.

Point 1: "What was the type of paper?"

Response 1: Calendered, uncoated, and made from a mix of bleached hardwood and softwood kraft pulps.

Revised version: "Calendered uncoated paper from industrial origin, produced from bleached wood kraft pulp, with an approximate grammage of 78 g/m2, was used in all coating experiments."

Point 2: line 44 and 80 - subscript

Response 2: Formulae corrected.

Point 3: Table 1. I suggest using abbrev. "Multi" instead of "Poly", the same changes in the text.

Response 2: Accordingly, "Poly" has been changed to "Multi" in the text, in Tables 1 and 6, and in Figure 6.

Point 4: Point 2.5. Due to lot of wrong calculations of WVTR in many articles I suggest adding equation for WVTR determination.

Response 4: Unlike for permeance, where pressure has to be taken into account, ASTM E96's equation is quite simple. It has been added to the revised version, as suggested. Moreover, we now clarify that we use the dry cup or dissecant method, out of the two methods that ASTM E96 prescribes (weight increase and weight decrease).

Point 5: Figure 1. Reference paper should be also presented here.

Response 5: It is: Uncoated in the legend, blue color.

Point 6: Page 6 - Were any differences in contact angle values for engraved and smooth roll? It should be included in the disscusion too.

Response 6: The revised version now says: "Differences between the two applicators were non-significant."

Point 7: Page 9 - roughness should be analytically confirmed via e.g. SEM (surface and cross-section)

Response 7: Following this recommendation, surface and cross-section micrographs have been aded and discussed in the new section, "3.1".

Regarding roughness, since the time we submitted the original version we have grown suspicious of our Bendtsen tester. Assays with samples of known roughness revealed that the systematic error was too high. Therefore, we have decided to take roughness measurements out of the manuscript, as they were not reliable. 

In turn, the breaking length has been determined for all samples, not only for multi-purpose ones.

Point 8: Line 297 - There is no information about polymer crustallinity, add some reference at least

Response 8: Accordingly, Assender & Windle's article has been cited.

Point 9: I suggest using therm "grammage" instead of "basis weight"

Response 9: This modification has been performed all along the manuscript, including the five tables — plus one additional table in the revised version.

Point 10: Due to the manuscript was submitted to "Nanomaterials" Journal role of nanocellulose should be highlighted in the study.

Response 10: This is the main reason why we decided to add a new section, "3.1", with its own table (isolating nanocellulose-only coatings), the aforementioned micrographs, and highlighting the role of nanocellulose. 

Text: "CNFs played at least four roles in coating processes for barrier properties: thickener, dispersion stabilizer..."